# A fairness assessment of mobility-based COVID-19 case prediction models

**Abdolmajid Erfani** [1] *, **Vanessa Frias-Martinez** [2,3]

**1** Department of Civil, Environmental, and Geospatial Engineering, Michigan Technological University, Houghton, MI, United States of America, **2** College of Information Studies, University of Maryland, College Park, MD, United States of America, **3** University of Maryland Institute for Advanced Computer Studies, University of Maryland, College Park, MD, United States of America

* aerfani@mtu.edu

## Abstract

In light of the outbreak of COVID-19, analyzing and measuring human mobility has become increasingly important. A wide range of studies have explored spatiotemporal trends over time, examined associations with other variables, evaluated non-pharmacologic interventions (NPIs), and predicted or simulated COVID-19 spread using mobility data. Despite the benefits of publicly available mobility data, a key question remains unanswered: are models using mobility data performing equitably across demographic groups? We hypothesize that bias in the mobility data used to train the predictive models might lead to unfairly less accurate predictions for certain demographic groups. To test our hypothesis, we applied two mobility-based COVID infection prediction models at the county level in the United States using SafeGraph data, and correlated model performance with sociodemographic traits. Findings revealed that there is a systematic bias in models' performance toward certain demographic characteristics. Specifically, the models tend to favor large, highly educated, wealthy, young, and urban counties. We hypothesize that the mobility data currently used by many predictive models tends to capture less information about older, poorer, less educated and people from rural regions, which in turn negatively impacts the accuracy of the COVID-19 prediction in these areas. Ultimately, this study points to the need of improved data collection and sampling approaches that allow for an accurate representation of the mobility patterns across demographic groups.

## Introduction

The interactions between human mobility and epidemic spread have been studied unprecedentedly during the COVID-19 pandemic [1–8]. With these efforts, nonpharmaceutical interventions (such as national lockdowns) have been evaluated for their effectiveness and socioeconomic impact on different groups [9–11], models have been developed to predict disease spatial diffusion [12, 13], and scenarios have been modeled to assess their outcomes [14–17]. Studies have demonstrated that mobility data are a meaningful proxy measure of social distancing [18], affect viral spreading [19, 20], and are useful for predicting the spread of COVID-19 [21–23].

**Data Availability Statement:** All data we use in this study are all publicly available. These datasets include the SafeGraph mobility data (https://docs.safegraph.com/docs/social-distancing-metrics), COVID-19 confirmed cases (https://github.com/

CSSEGISandData/COVID-19), and
sociodemographic information at US counties level
(https://www.census.gov/data/datasets.html).

**Funding:** The authors acknowledge funding
support from the National Science Foundation
under Grant Numbers NSF 1750102 and NSF
2210572. The views, opinions, conclusions, or
recommendations expressed in this research are
those of the authors and do not necessarily reflect
the view of the funding agencies.

**Competing interests:** The authors have declared
that no competing interests exist.

In particular, to control the spread of new cases and plan efficiently for hospital needs and capacities during an epidemic, public health decision-makers require accurate predictions of future case numbers [7]. For example, a study by Ilin et al. (2021) showed that changes in mobility can be used to predict COVID-19 cases. Their study demonstrated that public mobility data can be used to develop reduced-form and simple models that mimic the behavior of more sophisticated epidemiological models for predicting COVID-19 cases on a 10-day basis [21]. Another study examined several state-of-the-art machine learning models and statistical methods and demonstrated how mobility data can improve prediction trends when used as exogenous information in models [22].

As discussed, mobility data from anonymized smartphones has been shown to improve COVID-19 case prediction models. However, mobility data bias has received little attention in this predictive context. There exist only just a handful of papers reporting demographic bias in mobility data due to differences in smartphone ownership and use [24–26]; and since data providers are not transparent about how mobility data is collected, or about the socio-economic and demographic groups represented in them, directly measuring and correcting bias in mobility data is difficult [27]. In this study, we hypothesize that the presence of socio-economic and demographic bias in the mobility data used to train the COVID-19 case predictive models, might result in unfairly less accurate predictions for particular socio-economic and demographic groups. Unfair predictions provided to decision makers e.g., predictions of COVID-19 cases for minority groups that are lower than reality, could in turn be used to unfairly assign more resources to population groups that do not necessarily need them.

To test our hypothesis, we evaluated the performance of two types of mobility-based COVID-19 case prediction models highly used by decision makers due to its interpretability: linear regressions and time series models. In contrast to more complex epidemiological models that are hard to tune due to its parametric nature, and deep learning models that are difficult to interpret, linear models and time series are easy to train and test [21, 28–30]. The models were trained using SafeGraph's mobility data, and performance was measured via predictive errors. To assess the fairness of the predictions, we analyzed the relationship between the model prediction errors and specific socio-economic and demographic features at the county level in the United States and across the two model types. Evaluating the performance of two diverse interpretable models allowed us to account for potential algorithmic bias i.e., bias introduced by the algorithm itself [31, 32]. If unfair predictions are pervasive across types of models trained and tested with the same data, we can partially attribute the unfairness to the mobility data itself.

## Material and methods

In our study, we use mobility data from SafeGraph to build COVID-19 case prediction models; and we explore model performance across socio-economic and demographic features to potentially identify unfair results for specific groups i.e., differences in error distributions across social groups. We next describe these three types of datasets, with all being publicly available.

### Human mobility

We used SafeGraph's publicly-available human mobility data at the county level in the US. SafeGraph uses location information extracted from smartphones to provide aggregate data characterizing mobility in terms of visit volumes to types of places and volumes of origin-destination (OD) flows [33]. For this study specifically, we used the data publicly available in the origin-destination-time (ODT) platform [34], that computes OD flows between counties as

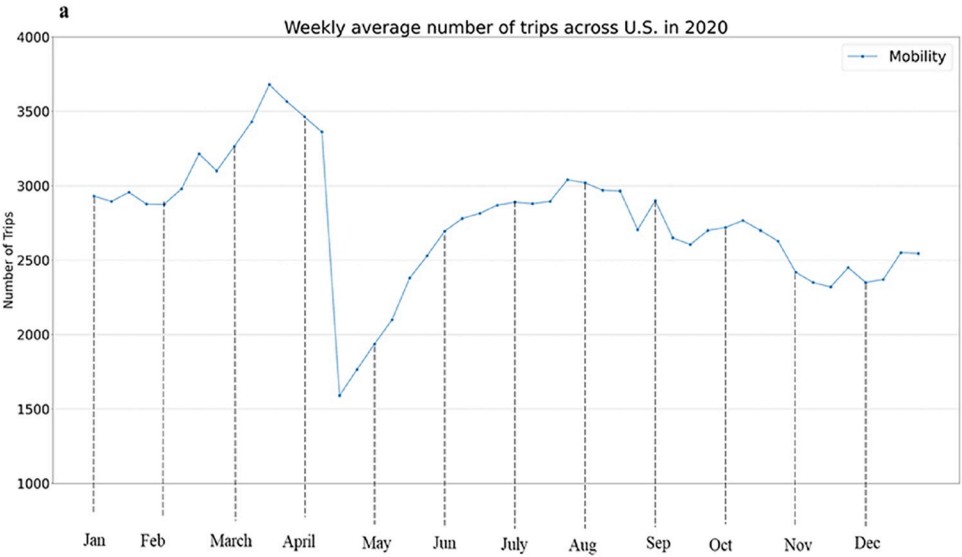

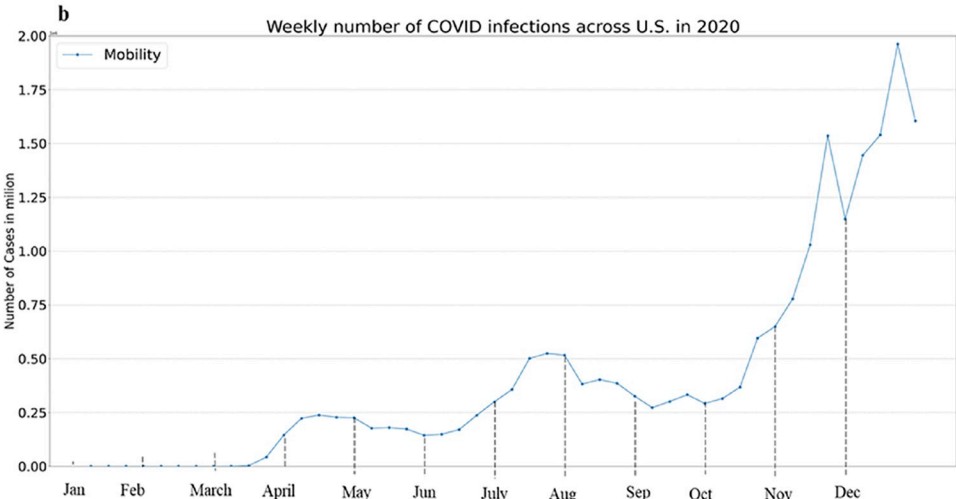

**Fig 1. Data on mobility measures, COVID-19 infections.** (a) Weekly average number of trips across the U.S. (b) Weekly new number of COVID infections across the U.S.

the aggregation of trips that start at an individual's home county location (origin), with a destination defined as a stay location within a county for longer than a minute. OD flows between all counties in the US were collected throughout all days of the year 2020. Fig 1A illustrates how the average number of trips at county level across the US changed over the year 2020. According to various studies in the US using mobility data, the dataset collected in Fig 1A also shows similar trends of mobility change [35, 36].

## COVID-19 cases

In order to obtain the cumulative and daily confirmed cases of COVID-19 for each county unit, we refer to the data repository compiled by the Johns Hopkins Center for Systems Science and Engineering (CSSE) [37]. As shown in Fig 1B, the weekly number of new COVID infections has been increasing over the year 2020.

**Table 1. Summary statistics of input variables at the county level.**

| Variable | Description | Count | Mean | Min | Max |
|---|---|---|---|---|---|
| Population | Population | 3,036 | 104,043.0 | 441 | 10,081,570 |
| Income | Median household income ($) | 3,036 | 70,264.8 | 35,819 | 181,261 |
| Education | Percentage of the population with a bachelor's degree and above (%) | 3,036 | 22.0 | 3.2 | 77.6 |
| Age | Median age | 3,036 | 41.3 | 22.3 | 67.4 |
| Smartphone | Percentage of the population who own smartphone (%) | 3,036 | 72.7 | 25.0 | 92.4 |
| NCHS | Urban-Rural Classification (1–6) | 3,036 | 4.9 | 1.0 | 9.0 |

### Socio-economic and demographic data

Data on socio-economic and demographic characteristics at the county level was also collected from public databases (US American Community Survey census data, ACS 2020) [38]. Studies have shown that sociodemographic factors such as age, race, income, educational level, and area of residence can influence smartphone ownership and usage, which may have an impact on mobility data biases [39, 40]. Therefore, we collected a wide range of information, including the population, income, education, age, and ownership of smartphones at the county level. Also, we used the US National Center for Health Statistics (NCHS) Urban-Rural Classification Scheme for Counties [41], which assigns each county an ordinal code ranging from 1 (most urban) to 6 (most rural). Table 1 summarizes the demographic features of the study with some descriptive statistics.

### Dataset preparation

We took the following steps to prepare the final dataset for modeling. Daily mobility data was collected from the origin-destination-time (ODT) platform [34] from April 14[th] to December 30[th]. The platform had daily mobility OD flows for 3,036 out of the 3,142 counties in the US. As a result, the total dataset size was of over 774,000 records. We used the Federal Information Processing Standard (FIPS) code, to match the daily number of COVID-19 cases per county with its corresponding mobility data. Therefore, the final dataset represented daily count of infections and mobility metrics per county in the US throughout the period of study. Following a similar procedure, we added socio-economic and demographic features at the county level to each data record using the FIPS code and the variables provided by the 5-year US ACS census from 2020 [38].

### Methods

In this section we will describe (i) the two types of models used in the COVID-19 case prediction; (ii) the training and evaluation of these models; and (iii) the process proposed to evaluate the fairness of the predictions across socio-economic and demographic groups, as well as across models.

### Models

**Model 1: Linear regression (Ilin et al. [21], Wang et al. [42], Ayan et al. [43], and Sahin [44]).** Several papers have suggested that linear regressions that combine mobility data with historical COVID-19 cases can successfully predict future cases [21, 42–44]. These models generally use different lags between mobility rates and COVID-19 cases to account for the infection period i.e., the period between the person's movement–and potentially interaction with others and infection–and the person testing positive for COVID-19. For this study we use Ilin

at al. linear model [21] because rather than picking one lag, they propose to consider multiple lags within the model encompassing the plethora of linear regressions that have been tested in the literature. Specifically, Ilin at al. (2021) use a distributed-lag model to estimate log confirmed infections as the dependent variable, with average mobility over lags 1–7, 8–14, and 15–21 days to predict the number of COVID-19 cases at a given day:

$$\log \frac{I_{it}}{I_{i,t-1}} = \beta_1 m_{1-7,it} + \beta_2 m_{8-14,it} + \beta_3 m_{15-21,it} + \epsilon_{it} \qquad (1)$$

where i is the unit of analysis, $\log \frac{I_{it}}{I_{i,t-1}}$ is the first difference of log confirmed cases at time t, $m_{1-7,it}$, $m_{8-14,it}$, $m_{15-21,it}$ represents mobility measures averaged over lags 1–7, 8–14 and 15–21, respectively, and $\beta_1$, $\beta_2$, $\beta_3$ are model parameters to be estimated.

**Model 2: Time series forecasting (Aji et al. [29], Zhao et al. [30], Zeng et al. [45], and Klein et al. [46]).**   The Autoregressive Integrated Moving Average (ARIMA) model is a statistical method that considers both past and present data for forecasting. An ARIMAX model, also known as ARIMA with multiple regressors, extends the basic ARIMA model to include other external variables for prediction. In the COVID-19 setting, mobility data and other sources of information have been used as regressors to potentially improve the predictive models [30, 45, 46]. For example, in their study Zhao et al. [30] conclude that with mobility data, time series forecasting provides accurate predictions with mobility data lags of between 8–10 days for dense or sparse populations respectively. In this study, we consider an ARIMAX (p, d, q) model that can be expressed as:

$$y_t = \beta_0 x_t + \sum_{j=1}^{p} \emptyset_j y_{t-j} + \varepsilon_t + \sum_{j=1}^{q} \theta_j \varepsilon_{t-j} \qquad (2)$$

where y is the number of confirmed infections, x is the mobility change as exogenous variable lagged by 21 days (similar to Model 1's selection of lag), p is the Autoregressive (AR) parameter, q is the Moving Average (MA) parameter, d is the degree of first differencing to make data stationarity, $\varepsilon$ is the error, and $\beta_0$, $\emptyset_j$, $\theta_j$ are model parameters to be estimated. By using the Python package Auto Arima, we were able to generate the best p, d, and q values based on the data set, thus providing better forecasts [47]. To summarize, the lag of mobility, historic number of COVID cases can be used to predict future cases at unit of analysis.

## Training and model evaluation

To train and evaluate the models, we used both historical COVID-19 data and mobility OD flows from mid-April to December 2020. Rather than using the raw mobility OD flows, we used a measure of mobility change over a baseline, which was calculated by dividing the daily mobility by the average daily mobility in February 2020, a non-holiday month before the COVID-19 pandemic. This is a common approach in prior COVID-19 case predictive models that use mobility data [21, 30].

The two models were trained at the county level on a daily basis using both COVID-19 case numbers and changes in mobility OD flows as independent variables to predict future cases. Socio-economic and demographic data were not used to train the models. That information was exclusively used during the fairness evaluation. For the linear regression model (Model 1), 21 days of mobility and past COVID-19 case data were used at a time for the training, and the trained model was used to test 1-day and 7-day predictions. We implemented a 1-day sliding window to replicate this train-test approach throughout the time period of analysis and reported average daily prediction error rates. Similarly, the time series model (Model 2), was

also trained using a typical training-testing window approach for time series predictions [48], with a 90-day training dataset. Using a 1-day sliding window on the training dataset, this approach resulted in predictions available from early August to the end of the year. Different training lengths were evaluated for both models, and the ones with the best accuracies were selected. In this process, thousands of regressions and ARIMAX models are trained at the county level on a daily basis to be able to predict COVID-19 cases. Once trained, each model was used to predict the number of COVID-19 cases for two lookaheads: 1-day (next day) and 7-days time (week) intervals at the county level, as predictions on a daily and weekly basis are a common theme in previous studies [21, 29, 30, 42–45].

Finally, the model performance was evaluated via the error rate, which was calculated on a daily basis based on the difference between the actual number of COVID cases and predictions as Eq 3. A mean absolute percentage error rate (MAPE) is calculated by averaging the error rates for specific counties over a given time period.

$$Error\ rate_t = \Big|\frac{Prediction\ value_t - Actual\ number\ of\ COVID\ cases_t}{Actual\ number\ of\ COVID\ cases_t}\Big| \tag{3}$$

$$MAPE = \frac{100\%}{n}\sum_{t=1}^{n} Error\ rate_t \tag{4}$$

## Fairness analysis

We analyzed the fairness of the predictions for each model by computing the weekly MAPE per lookahead (1-day and 7-day) at the county level, followed by a spearman rank-order correlation analysis between the average weekly error rate across counties in the US and their socio-economic and demographic characteristics presented in the data section: household income (average household income), smartphone ownership (percentage of households owning smartphones), population, education level (bachelor's degree), urbanity-rurality level (NCHS classification), and age (median age). A spearman correlation provides an opportunity to investigate the monotonic relationship between two continuous variables of demographic features and model accuracy. A monotonic relationship occurs when the variables change together, but not necessarily at the same rate [49, 50]. Using the P-value to evaluate the correlation analysis significance, we can assess whether performance is similar (fair) or not (unfair) between social groups. To discuss correlation strength, and based on prior work, we will use 0.3 as the correlation coefficient threshold between a high and low correlation [51], or a weak and a moderate correlation [52].

## Results

### Model performance

First, we discuss the COVID-19 case prediction performance of the two models presented (see Fig 2).

As we can observe, both models predict the number of next day cases (1-day) with an average weekly error rate of 10–20%, and the number of cases in a week (7-days) with an average weekly error rate of 30–40%. Models' performance is in a comparable range to previous studies [21, 29, 30, 42–45], but with the difference that we reported the results for the entire year of 2020 and the US, not specific regions or COVID waves.

### Fairness performance

For Model 1, we observe a negative and statistically significant spearman rank correlation between the prediction error rate and income (R (1-day) = -0.13, R (7-days) = -0.08, p-

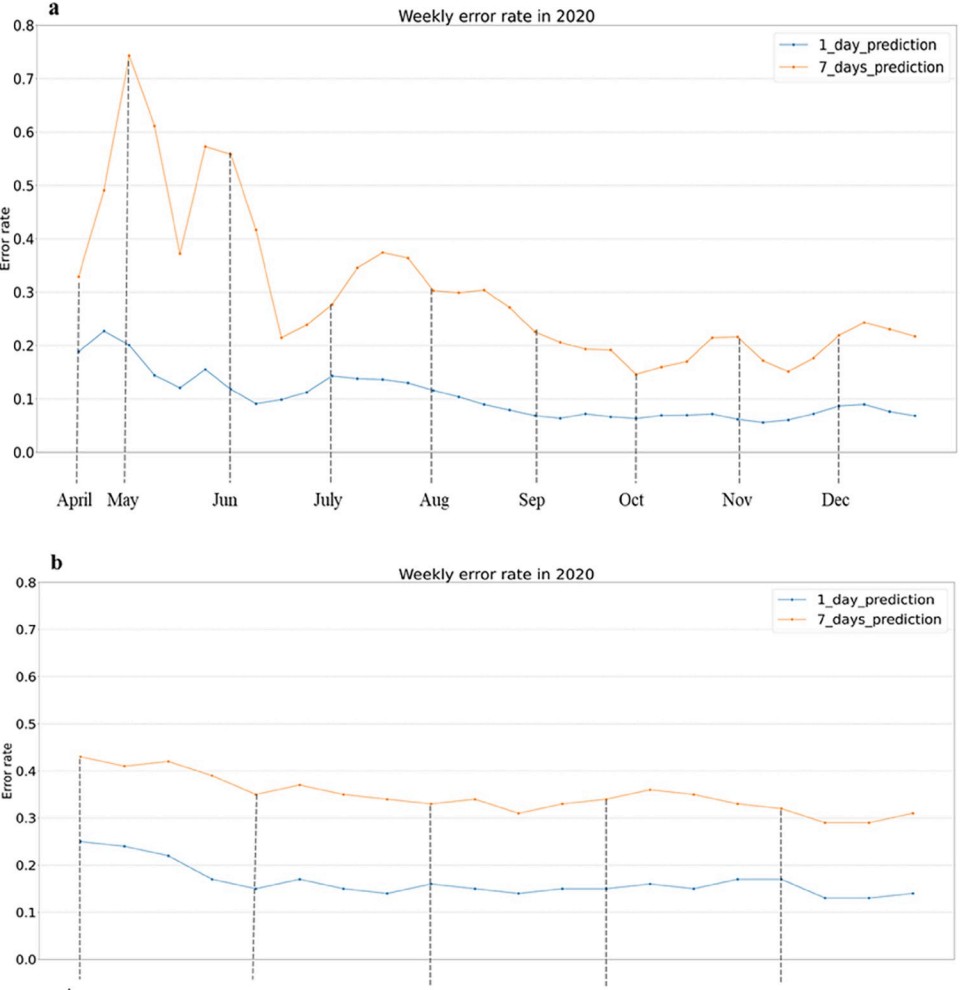

**Fig 2. Prediction error rate on a weekly basis.** (a) Regression model (b) Time series model.

value < 0.001), smartphone ownership (R (1-day) = -0.14, R (7-days) = -0.09, p-value < 0.001), population (R (1-day) = -0.11, R (7-days) = -0.07, p-value < 0.001), bachelor degree (R (1-day) = -0.13, R (7-days) = -0.09, p-value < 0.001). The results suggest that Model 1 –a regression model of COVID-19 cases with mobility–performs better (has fewer errors) in counties with higher incomes, higher smartphone ownership, larger populations, and higher educational levels. On the other hand, correlation analysis indicates a weak and positive relationship between NCHS code and error rate (R (1-day) = 0.21, R (7-days) = 0.15, p-value < 0.001) and median age (R (1-day) = 0.12, R (7-days) = 0.09, p-value < 0.01). Therefore, as rurality, and age increased, the model's error rate increased, suggesting it performs worse in rural areas and among older communities (Fig 3 represents the weekly correlations for some of these features).

For Model 2, we observe a negative and statistically significant spearman rank correlation between the prediction error rate and the income (R (1-day) = -0.13, R (7-days) = -0.09, p-value < 0.001), smartphone ownership (R (1-day) = -0.13, R (7-days) = -0.10, p-value < 0.001), population (R (1-day) = -0.11, R (7-days) = -0.11, p-value < 0.001), bachelor degree (R (1-day) = -0.13, R (7-days) = -0.08, p-value < 0.001). These results show that Model

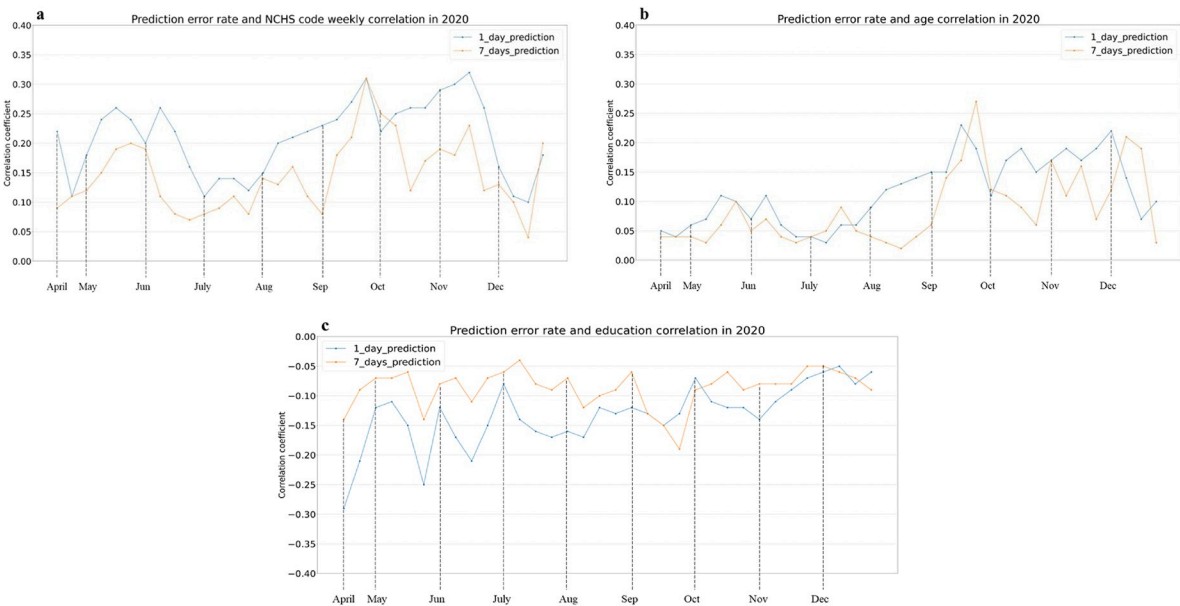

**Fig 3. Correlation analysis with selected factors on a weekly basis for Model 1.** (a) NCHS code (b) Age, and (c) education.

2 –an ARIMAX with mobility data as an exogenous variable–performs better (i.e., with lower errors) in counties whose income, smartphone ownership, population, and educational levels are higher. Fig 4 shows weekly correlations for some of these features. On the other hand, the correlation analysis also reveals a weak and positive relationship between the error rate and the NCHS code (R (1-day) = 0.20, R (7-days) = 0.21, p-value < 0.001) and median age (R (1-day) = 0.08, R (7-days) = 0.09, p-value < 0.01). In other words, the model's error rate increased as

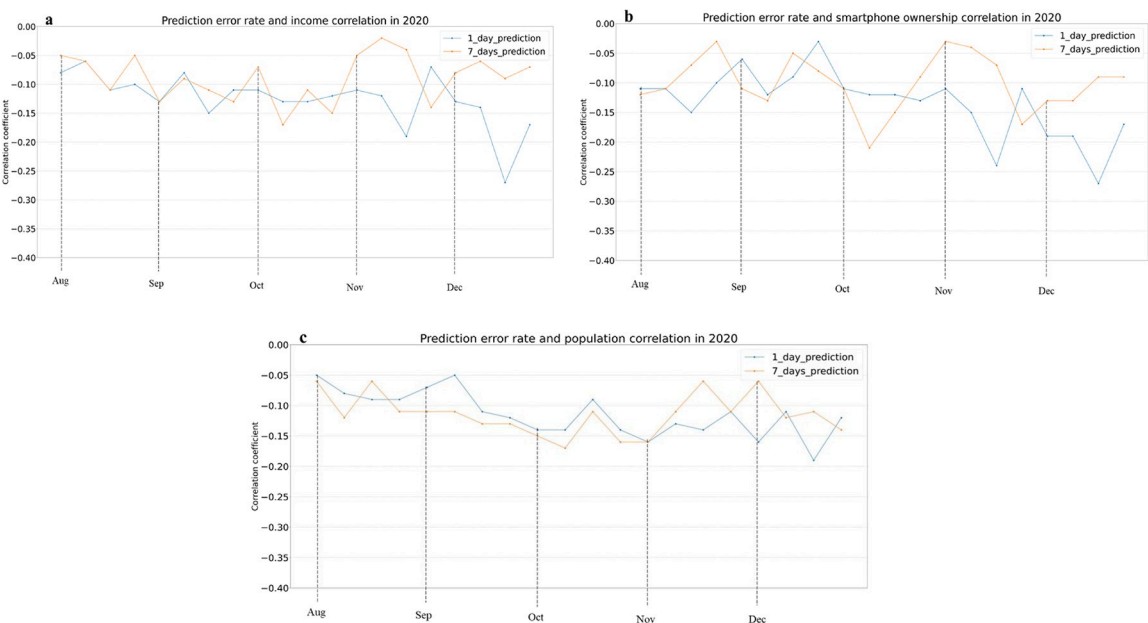

**Fig 4. Correlation analysis with selected factors on a weekly basis for Model 2.** (a) Income (b) Smartphone ownership, and (c) Population.

**Table 2. County-level correlations between Model 1 error rate and sociodemographic features.** (Note: Statistical significance: *** p_value < 0.001, ** p_value < 0.01, * p_value < 0.05).

| | Income | | Smartphone | | Population | | Education | | NCHS | | Age | |
|---|---|---|---|---|---|---|---|---|---|---|---|---|
| | 1 day | 7 day | 1 day | 7 day | 1 day | 7 day | 1 day | 7 day | 1 day | 7 day | 1 day | 7 day |
| **April** | -0.20*** | -0.09*** | -0.19*** | -0.09*** | -0.14*** | -0.07*** | -0.25*** | -0.12*** | 0.17*** | 0.11*** | 0.05* | 0.04* |
| **May** | -0.14*** | -0.06*** | -0.16*** | -0.07*** | -0.14*** | -0.09*** | -0.16*** | -0.09*** | 0.23*** | 0.17*** | 0.08*** | 0.03** |
| **Jun** | -0.14*** | -0.09*** | -0.14*** | -0.07*** | -0.09*** | -0.05*** | -0.16*** | -0.08*** | 0.21*** | 0.11*** | 0.07*** | 0.05*** |
| **July** | -0.14*** | -0.07*** | -0.08*** | -0.06*** | -0.07*** | -0.05*** | -0.14*** | -0.07*** | 0.13*** | 0.09*** | 0.05* | 0.06*** |
| **August** | -0.15*** | -0.07*** | -0.15*** | -0.08*** | -0.11*** | -0.05*** | -0.15*** | -0.10*** | 0.20*** | 0.14*** | 0.12*** | 0.03* |
| **September** | -0.12*** | -0.12*** | -0.19*** | -0.17*** | -0.13*** | -0.10*** | -0.13*** | -0.13*** | 0.26*** | 0.20*** | 0.18*** | 0.16*** |
| **October** | -0.10*** | -0.11*** | -0.16*** | -0.14*** | -0.14*** | -0.11*** | -0.11*** | -0.10*** | 0.26*** | 0.22*** | 0.16*** | 0.13*** |
| **November** | -0.11*** | -0.07*** | -0.17*** | -0.09*** | -0.11*** | -0.07*** | -0.10*** | -0.07*** | 0.29*** | 0.18*** | 0.18*** | 0.13*** |
| **December** | -0.09*** | -0.06*** | -0.09*** | -0.09*** | -0.07*** | -0.06*** | -0.06*** | -0.07*** | 0.14*** | 0.12*** | 0.13*** | 0.14*** |

rurality and age increased, revealing a model that performs worse in rural environments, and among older populations. Due to the replication of these findings in models 1 and 2, which controls for algorithmic bias, we posit that this model is unfair in part because it uses biased mobility data, although bias in the way COVID-19 case data is gathered (e.g., under-reporting) could also influence its outcome.

To summarize the fairness analysis across models, Tables 2 and 3 provide the monthly correlation averages between the sociodemographic factors at the county level and the error rates for Models 1 and 2 1-day and 7-day predictions, respectively. As discussed, due to the diverse size of the optimal training windows, Model 1 predictions run from April till December, while Model 2 predictions are produced from August till December. With a few fluctuations, and as discussed in the weekly analyses in Figs 3 and 4, both models show the same pattern of results throughout 2020: lower prediction errors in large, highly educated, wealthy, young, and urban counties. Given that the strength of the correlations found is weak, we posit that all socio-economic and demographic features are related to significant, albeit weak, bias. We do not observe statistically significant differences in the strengths across features. We do however see that correlation coefficients are smaller for 7-day predictions than 1-day predictions, which might point to more negligible bias for these models. Nevertheless, the statistical results do not support any feature being more biased than other. In addition, it is important to highlight that for certain socio-economic and demographic features in Tables 2 and 3, we observe some variance in the correlation coefficients across months, with earlier pandemic months showing higher coefficients. We posit that these might be due to mobility behaviors being more entropic later during the pandemic, which might make it harder to find associations. As prior work has shown, uniform mobility behaviors made mobility data more useful in predictive models at the onset of the pandemic than in later periods [20, 53].

**Table 3. County-level correlations between Model 2 error rate and sociodemographic features.** (Note: Statistical significance: *** p_value < 0.001, ** p_value < 0.01, * p_value < 0.05).

| | Income | | Smartphone | | Population | | Education | | NCHS | | Age | |
|---|---|---|---|---|---|---|---|---|---|---|---|---|
| | 1 day | 7 day | 1 day | 7 day | 1 day | 7 day | 1 day | 7 day | 1 day | 7 day | 1 day | 7 day |
| **August** | -0.09* | -0.07* | -0.12** | -0.08* | -0.08* | -0.09** | -0.10** | -0.06 | 0.14** | 0.15*** | 0.08* | 0.08** |
| **September** | -0.12*** | -0.12** | -0.08* | -0.09* | -0.09** | -0.12** | -0.14** | -0.09** | 0.15** | 0.16*** | 0.05* | 0.05* |
| **October** | -0.12** | -0.13*** | -0.12*** | -0.14*** | -0.13** | -0.15*** | -0.11*** | -0.09** | 0.19*** | 0.23*** | 0.08* | 0.08** |
| **November** | -0.12*** | -0.06* | -0.15*** | -0.08* | -0.14*** | -0.11*** | -0.14*** | -0.07* | 0.26*** | 0.31*** | 0.11* | 0.16** |
| **December** | -0.18*** | -0.08** | -0.21*** | -0.11** | -0.15*** | -0.11*** | -0.17*** | -0.08*** | 0.27*** | 0.22*** | 0.09** | 0.09** |

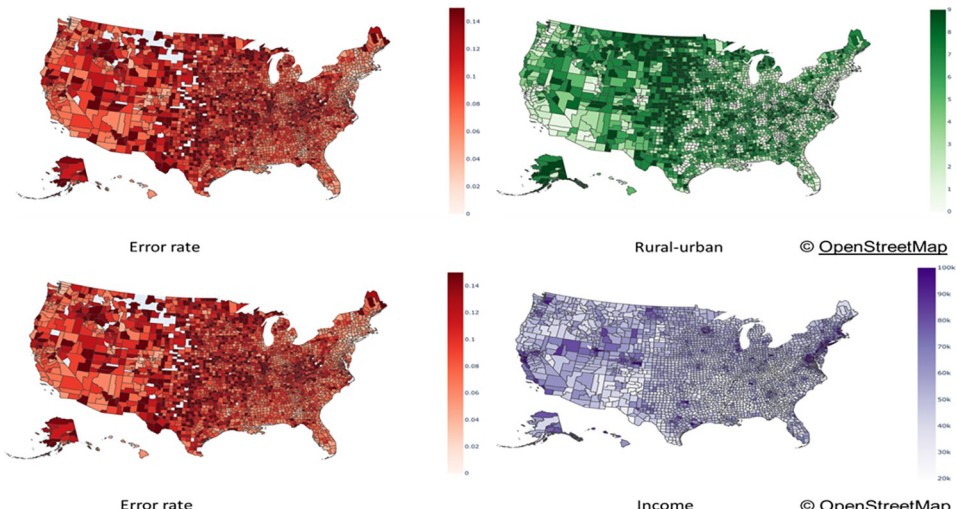

**Fig 5. Spatial comparison of error rate and demographic features for Model 1.** (These plots have been generated with Plotly open source graphing libraries using base maps from OpenStreetMap. OpenStreetMap is open data, licensed under the Open Data Commons Open Database License (ODbL) by the OpenStreetMap Foundation (OSMF)).

Fig 5 shows two comparative visualizations between the average 1-day prediction error rate for the regression model (Model 1) and two demographic features namely urbanity level and household income. Visualizations for the other demographic features are shown in the supplemental materials, S1 Fig in S1 File. The visualizations show trends in line with the quantitative results discussed before i.e., that areas with a higher rurality (dark green) and areas with a lower income rate (white) have a higher error rate (dark red). We observe two interesting patterns. First, the error rates are much higher across the eastern states of the Great Plains (vertical line from North Dakota to Texas) which represent some of the highest categories of rurality and some of the lowest income rates (outside of metropolitan areas in the region). Second, the error rates are higher in the Appalachian region (from southern New York to northern Mississippi) which is associated to some of the lowest income ratios in the country. Similar visualizations were observed for 1-day predictions for Model 2. Visualizations for 7-day predictions did not show clear spatial trends, possibly due to the weaker correlations reported. Map figures are generated using the Plotly package, an open source Python package.

## Discussion

To combat the COVID-19 pandemic, governments and private companies around the world were promoting the use of digital public health technologies for data collection and processing [54–58]. Through the use of GPS, cellular networks and Wi-Fi, smartphones can collect and aggregate location data in real-time to monitor population flows, identify transmission hotspots, determine the effectiveness of non-pharmacologic interventions [59], and predict future COVID-19 cases [7, 20, 25].

Using SafeGraph's mobility data, we examined whether two popular predictive models that use mobility data to predict COVID-19 cases over time, performed fairly across social groups. Our findings revealed a correlation between a county's socio-economic and demographic characteristics and the models' error rates. In particular, we observed that the prediction errors were lower in large, highly educated, wealthy, young, and urban counties. Given that the findings were similar across models, thus controlling for algorithmic bias, we posit that the

presence of bias in the mobility data negatively impacts the model predictions by unfairly out-putting case numbers with higher errors for specific social groups. Furthermore, our results show that mobility data appears to be less likely to capture older, poorer, and less educated users. Thus, allocating public health resources based on such mobility data could disproportionately harm seniors and minorities at high risk.

For both predictive models, we observed higher biases in 1-day predictions compared to 7-day predictions. We posit this is possibly due to the already reported difficulty in predicting COVID-19 cases for higher lookaheads [60], thus resulting in noisier predictions which might in turn generate lower correlations between model performance and sociodemographic characteristics. When comparing the regression and time series models in terms of their biases, we did not observe any statistically significant difference between the reported correlation coefficients. A Wilcoxon rank sum test, also known as a Mann-Whitney U test [61], was implemented to compare two independent groups of coefficients on a 1-day prediction period (Tables 2 and 3). We chose this test because it is a non-parametric statistical test that is not based on assumptions of normality or equal variance. With a P-value of 0.289, the null hypothesis was accepted pointing to no significant difference between bias coefficients between the two models.

To generalize smartphone-derived insights over a population, the mobility data must reveal information about the population without bias i.e., information that is representative across socio-economic and demographic groups. However, due to the lack of ground truth data about the socio-economic and demographic characteristics of the population whose mobility data is collected, this study has also shown that investigating performance fairness can provide valuable insights into potential mobility data biases.

Finally, as the research community moves forward with the use of mobility data in COVID-19 case prediction models, we think it is important to consider the following set of recommendations. First, and whenever possible, we strongly suggest applying sampling bias mitigation approaches to correct for under-represented groups in the data, as prior work has successfully done [48, 62, 63]. Second, mitigation approaches might not always be possible, due to the lack of demographic information about the individuals whose cell phone data is being collected. For that reason, we encourage the research community working with mobility data to report fairness analyses together with the performance of the predictive models proposed. We hope that these practices will enhance pandemic management via case prediction models that are more transparent and fairer, and that will allow for more equitable decision making.

## Limitations

While this study addressed potential biases in mobility data currently used by two types of predictive models, there are a number of limitations related to modeling and dataset biases that require clarification.

We chose a baseline period of one pre-covid month in 2020 to model 'normal behavior'. This choice was determined by the limited availability of "free" mobility data. Although, ideally, mobility baselines should be from a pre-covid period e.g., 2019, we were limited by the availability of free SafeGraph data, which started in 2020. Testing different baselines is an important research question, but that would require having access to additional mobility data that was not available.

Other limitations include the decision to add mobility data in the predictive models using mobility changes from up to 21 days prior to the prediction date. Although earlier mobility periods could be considered, the probability that mobility patterns prior to 21 days might translate into a COVID-19 infection is extremely low given that the incubation periods known

for COVID-19 and its variants can be up to 14 days [64, 65]. Prediction results are only reported for 1 and 7 days ahead despite the fact that different testing lookaheads might provide diverse outcomes.

We have reported fairness analysis results in terms of correlation coefficients between performance and socio-economic variables. Nevertheless, statistically significant correlations reflect the probability of such a correlation occurring rather than its strength. Correlation coefficient strengths can be interpreted differently across scientific fields, and authors should avoid overinterpreting associations [51, 66, 67]. Based on prior work utilizing spearman rank correlation in the context of medicine and big data analysis, we have selected a correlation coefficient of 0.3 as the threshold between high and low correlation [51], or weak and moderate correlation [52]. As a final point, in addition to mobility data, there are other sources of data that might include biases and have an effect on prediction performance, such as diverse COVID-19 case reporting methodologies and US census tract data, both of which are used in this paper. It is important to note that these are potential biases in our study, and future work should look into their effect on the fairness analysis presented in this paper.

Finally, we made several conscious modeling choices. First, we focused on linear regressions and time series models due to its interpretability. In contrast with more complex epidemiological models that are hard to tune due to its parametric nature, and deep learning models that are difficult to interpret, linear models and time series are easy to train and test [21, 28–30]. Second, we have avoided incorporating socio-economic and demographic variables as input to the linear regression and time series prediction models. This choice was based on prior work showing that the addition of demographic features as input to predictive models is not only controversial but also potentially harmful [68]. In fact, it has been argued that using socio-economic or demographic data as predictor may instead reinforce bias and generate predictions based primarily on demographic variables rather than on more actionable parameters, thus perpetuating inequalities [69]. Future studies could consider incorporating demographic features as inputs to the predictive models to replicate the fairness analysis presented in this paper. Third, we trained individual prediction models per county. Future work should explore a unified model that learns COVID-19 trends for all counties. This approach would allow for inclusion of county-level variables indicative of population vulnerability directly in the model, potentially yielding more accurate results.

## Supporting information

**S1 File.**
(DOCX)

## Author Contributions

**Conceptualization:** Abdolmajid Erfani, Vanessa Frias-Martinez.

**Data curation:** Abdolmajid Erfani.

**Formal analysis:** Abdolmajid Erfani.

**Methodology:** Abdolmajid Erfani, Vanessa Frias-Martinez.

**Project administration:** Vanessa Frias-Martinez.

**Resources:** Vanessa Frias-Martinez.

**Supervision:** Vanessa Frias-Martinez.

**Validation:** Vanessa Frias-Martinez.

**Visualization:** Abdolmajid Erfani.

**Writing – original draft:** Abdolmajid Erfani.

**Writing – review & editing:** Abdolmajid Erfani, Vanessa Frias-Martinez.

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
