## [Decision Letter · Decision Letter 0]

22 May 2023

PONE-D-23-04556A fairness assessment of mobility-based COVID-19 case prediction modelsPLOS ONE

Dear Dr. Erfani,

Thank you for submitting your manuscript to PLOS ONE. After careful consideration, we feel that it has merit but does not fully meet PLOS ONE’s publication criteria as it currently stands. Therefore, we invite you to submit a revised version of the manuscript that addresses the points raised during the review process.

We look forward to receiving your revised manuscript.

Kind regards,

Emanuele Crisostomi, PhD

Academic Editor

PLOS ONE

Journal Requirements:

Reviewers' comments:

Reviewer's Responses to Questions

**Comments to the Author**

1. Is the manuscript technically sound, and do the data support the conclusions?

Reviewer #1: Yes

Reviewer #2: Partly

2. Has the statistical analysis been performed appropriately and rigorously? 

Reviewer #1: Yes

Reviewer #2: Yes

3. Have the authors made all data underlying the findings in their manuscript fully available?

Reviewer #1: Yes

Reviewer #2: Yes

4. Is the manuscript presented in an intelligible fashion and written in standard English?

Reviewer #1: Yes

Reviewer #2: Yes

5. Review Comments to the Author

Reviewer #1: I commend the authors for their research in evaluating the performance of mobility-based COVID-19 prediction models and the consideration of equity in prediction across different demographic groups. The results of the study are important to recognize potential biases in these prediction models and apply fair assessment to vulnerable populations. I recommend major revision before accepting this journal as there is a lack of interpretability of the results and the limitations of the study itself. Please see my comments below:

1) Certain statements in the research article are missing the proper citations. For example, “since data providers are not transparent about how mobility data is collected, or about the socio-economic and demographic groups represented in them, directly measuring and correcting bias in mobility data is difficult” could use supporting literature. Another example is “First, and whenever possible, we strongly suggest applying sampling bias mitigation approaches to correct for under-represented groups in the data” which could reference successful examples of sampling bias mitigation.

2) Greater explanation is needed in the socio-economic and demographic information. Although the authors mention the names of the demographic variables, what are the exact values and/or thresholds used in the prediction models and correlation analysis? For example, is education made of multiple categories (high school, associates, bachelors, graduate) or a binary category (below vs. higher than high school)? This could be explained in the text or summarized in a table.

3) The authors mention that the two-week period is a common approach. However, for the sake of this analysis, have the authors performed a sensitivity analysis of the baseline period? Choosing the right baseline could influence the performance of the prediction models. At the very least, this should be mentioned as a potential limitation in the study.

4) It is hard to understand the results from a spatial perspective. The authors could map the US counties with the different error prediction rates and associated demographics for a specific time period. This could lead to some interesting discussion about how error prediction rates are connected to areas of vulnerable demographics and spatial biases.

5) The research paper mentions little of their own limitations which could affect the replicability of the study and reliability of the results. These limitations could be made clear to the reader in the discussion section. For example, although the correlations were statistically significant, certain correlations were extremely small, even below 0.1. A statistically significant correlation speaks more that the probability of such a correlation happening rather than the strength of the correlation between variables. This should be clarified to the reader. Another limitation is the potential bias in using US census data to represent demographic information. The authors did mention that COVID-19 data collection process could be biased, and this problem also extends to US census data.

6) The interpretability of the results could be expanded. Is there a reason why the 1-day models showed a higher bias that the 7-day models? Is there a significant difference in bias between the linear regression vs. the time series forecasting?

Reviewer #2: The paper reviews a prediction of COVID-19 cases based only on mobility data and raises the question of whether these predictions are reliable, considering the demographic differences among the population.

The analysis is done in two stages, first, the authors run a linear regression and a time-series forecasting to predict the number of COVID-19 cases. Then, they apply a Spearman correlation analysis between the errors of the model and socio-economic and demographic characteristics. The authors concluded that in bigger regions, with highly educated, wealthy, young, and urban areas the prediction errors are smaller.

Although, in general, the paper is easy to read, some parts could definitely be improved and clarified.

The main concern is the methodology choice, also, considering the low resulting correlation, the validity of the conclusions is a bit questionable. An alternative may be a choice of more sophisticated models for COVID-19 spread (for example, the ones which directly include socio-economic characteristics), to further compare their prediction errors with the regressions suggested by authors.

Some further comments are listed below:

1) The resulting correlation values look quite low. Could you please elaborate on the validity of the conclusions in the studied context?

2) The preparation of the datasets for the models is not clear. In particular, there is no information about the size of the dataset, its granularity, and the shares of the training and testing dataset. Also, it would be interesting to see the number of counties and their size.

3) It is not clear how the input of linear regression was formed – were all counties considered together or the coefficients were calculated separately for each county? It would be also convenient to summarize the explanatory variables fed to each model.

4) The abbreviations (for example, NCHS at line 192) and variable in equations should be defined.

6. PLOS authors have the option to publish the peer review history of their article (what does this mean?). If published, this will include your full peer review and any attached files.

Reviewer #1: No

Reviewer #2: No

---

## [Author Response · Author response to Decision Letter 0]

5 Jul 2023

The authors are grateful for the opportunity to resubmit the article. Please accept our thanks for the time and effort that the editors and reviewers spent reviewing our manuscript. All comments were seriously considered, and we conducted a thorough revision of the entire paper according to reviewers’ feedback. We hope the changes listed have made the manuscript suitable for publication and we look forward to your response. 

Please note that due to the changes, line and page numbers referenced in the reviewers’ comments might be different compared to the updated version. Where applicable, page and line numbers are listed based on the new version which can be tracked.

Editor: 

1) Comment: “Thank you for submitting your manuscript to PLOS ONE. After careful consideration, we feel that it has merit but does not fully meet PLOS ONE’s publication criteria as it currently stands. Therefore, we invite you to submit a revised version of the manuscript that addresses the points raised during the review process.”

• Response: We appreciate the time spent reviewing our paper by department editor, and reviewers. Our paper has been revised entirely in response to your feedback. Please see below for our detailed responses to each particular comment. 

Reviewer 1: 

2) Comment: “I commend the authors for their research in evaluating the performance of mobility-based COVID-19 prediction models and the consideration of equity in prediction across different demographic groups. The results of the study are important to recognize potential biases in these prediction models and apply fair assessment to vulnerable populations. I recommend major revision before accepting this journal as there is a lack of interpretability of the results and the limitations of the study itself. Please see my comments below:”

• Response: We appreciate your positive feedback. We hope the changes we have made have enhanced the interpretability of the paper. 

3) Comment: “Certain statements in the research article are missing the proper citations. For example, “since data providers are not transparent about how mobility data is collected, or about the socio-economic and demographic groups represented in them, directly measuring and correcting bias in mobility data is difficult” could use supporting literature. Another example is “First, and whenever possible, we strongly suggest applying sampling bias mitigation approaches to correct for under-represented groups in the data” which could reference successful examples of sampling bias mitigation.”

• Response: Thanks for your comment. We have added supporting citations for the statements that you highlighted and changed the text accordingly. 

- Grantz, K. H., Meredith, H. R., Cummings, D. A., Metcalf, C. J. E., Grenfell, B. T., Giles, J. R., ... & Wesolowski, A. (2020). The use of mobile phone data to inform analysis of COVID-19 pandemic epidemiology. Nature communications, 11(1), 4961.

- Griffin, G. P., Mulhall, M., Simek, C., & Riggs, W. W. (2020). Mitigating bias in big data for transportation. Journal of Big Data Analytics in Transportation, 2(1), 49-59.

- Garber, M. D., Labgold, K., & Kramer, M. R. (2022). On selection bias in comparison measures of smartphone-generated population mobility: an illustration of no-bias conditions with a commercial data source. Annals of Epidemiology, 70, 16-22.

- Yabe, Takahiro, Bernardo García Bulle Bueno, Xiaowen Dong, Alex Pentland, and Esteban Moro. "Behavioral changes during the COVID-19 pandemic decreased income diversity of urban encounters." Nature Communications 14, no. 1 (2023): 2310.

4) Comment: “Greater explanation is needed in the socio-economic and demographic information. Although the authors mention the names of the demographic variables, what are the exact values and/or thresholds used in the prediction models and correlation analysis? For example, is education made of multiple categories (high school, associates, bachelors, graduate) or a binary category (below vs. higher than high school)? This could be explained in the text or summarized in a table.

• Response: Thank you for your comment. We have added Table 1 to summarize the demographic features of the study and some descriptive statistics like feature description, count, mean, min, and max values. As an example, education is captured in this study by the percentage of the population with a bachelor's degree or higher at the county level.

Variable Description Count Mean Min Max

Population Population at county level 3,036 104,043.0 441 10,081,570

Income Median household income at county level 3,036 70,264.8 35,819 181,261

Education Percentage of population at county level with a bachelor degree and above 3,036 22.0 3.2 77.6

Age Median age at county level 3,036 41.3 22.3 67.4

Smartphone Percentage of population at county level with a bachelor degree and above 3,036 72.7 25.0 92.4

NCHS Urban-Rural Classification Scheme 3,036 4.9 1.0 9.0

Table 1. Summary statistics of input variables

5) Comment: “The authors mention that the two-week period is a common approach. However, for the sake of this analysis, have the authors performed a sensitivity analysis of the baseline period? Choosing the right baseline could influence the performance of the prediction models. At the very least, this should be mentioned as a potential limitation in the study.”

• Response: Thank you very much for your comment. We understand your concern regarding the use of “common approaches” in the related literature and their impact on model performance and fairness. There are, in fact, several modeling decisions that we have made which could influence the performance metrics we report. The baseline period of one pre-covid month in 2020 to model “normal behavior’ is determined by the limited availability of “free” mobility data. Although ideally mobility baselines should be from a pre-covid period e.g., 2019, we were limited by the availability of SafeGraph data, which started in 2020. Testing different baselines is an important research question, but that would require having access to additional mobility data that was not available to us. Nevertheless, we acknowledge this is a limitation and we have created a new limitations section where we discuss this, and other limitations.

Other limitations include the decision to add mobility data in the predictive models using mobility changes from up to 21 days prior to the prediction date. Although earlier mobility periods could be considered, the probability that mobility patterns prior to 21 days might translate into a covid-19 infection is extremely low given that the incubation periods known for covid-19 and its variants can be up to 14 days [1,2]. Finally, prediction results are only reported for 1 and 7 days ahead despite the fact that different testing lookaheads might provide diverse outcomes. Overall, we agree with the reviewer that all of the above are limitations of our paper, and that the results discussed need to be framed within these limitations. We have further clarified this in the new limitations section. Please refer to revision in lines 323-333, and also to the text below.

“We chose a baseline period of one pre-covid month in 2020 to model ‘normal behavior’. This choice was determined by the limited availability of “free” mobility data. Although, ideally, mobility baselines should be from a pre-covid period e.g., 2019, we were limited by the availability of free SafeGraph data, which started in 2020. Testing different baselines is an important research question, but that would require having access to additional mobility data that was not available. 

Other limitations include the decision to add mobility data in the predictive models using mobility changes from up to 21 days prior to the prediction date. Although earlier mobility periods could be considered, the probability that mobility patterns prior to 21 days might translate into a COVID-19 infection is extremely low given that the incubation periods known for COVID-19 and its variants can be up to 14 days 60,61. Prediction results are only reported for 1 and 7 days ahead despite the fact that different testing lookaheads might provide diverse outcomes.”

[60] https://www.webmd.com/covid/coronavirus-incubation-period

[61]https://www.pfizer.com/news/articles/why_the_covid_19_incubation_period_changes_and_how_that_can_affect_us

6) Comment: “It is hard to understand the results from a spatial perspective. The authors could map the US counties with the different error prediction rates and associated demographics for a specific time period. This could lead to some interesting discussion about how error prediction rates are connected to areas of vulnerable demographics and spatial biases.”

Response: Thanks for the constructive feedback, we agree with the reviewer that maps can offer a complementary approach to the analysis presented given its spatial nature. We have created choropleth maps of the US to show the distribution of the prediction error rates as well as the distribution of the socio-economic and demographic features. To be able to compare them, we have created pairs of plots where the left-hand side represents the error rate distribution and the right-hand side a specific socio-economic or demographic metric. Since errors are computed monthly, we have represented the average error across all months in the period under analysis. The value-ranges for the color-coded socio-economic and demographic features in the choropleths were selected based on best trend visualizations. In the plots below, we show the visualizations for the 1-day regression model (Model 1), although similar visualizations were obtained for the 1-day time series model (Model 2). In the paper, we have included two of these plots and a discussion highlighting the main spatial trends. A visual exploration confirms the quantitative results discussed in the paper i.e., that areas with a higher rurality (dark green), lower income rates (white), lower education rates (white), older population rates (dark orange), lower population rates (white) and lower smartphone ownership rates (white), tend to be associated to higher error rates (darker red). The plots also show two interesting patterns. First, the error rates are much higher across the eastern states of the Great Plains (vertical line from North Dakota to Texas) which represent some of the highest categories of rurality and older age rates as well as some of the lowest income and education rates (outside of metropolitan areas in the region). Second, the error rates are higher in the Appalachian region (from southern New York to northern Mississippi) which is associated to some of the lowest income and education rates in the country. Similar visualizations were observed for 1-day predictions for Model 2. Visualizations for 7-day predictions did not show clear spatial trends, possibly due to the weaker correlations reported. Please refer to revision in lines 254-264, and also to the text below.

“Figure 5 shows two comparative visualizations between the average 1-day prediction error rate for the regression model and two demographic features namely urbanity level and household income. The visualizations show trends in line with the quantitative results discussed before i.e., that areas with a higher rurality (dark green) and areas with a lower income rate (white) have a higher error rate (dark red). We observe two interesting patterns. First, the error rates are much higher across the eastern states of the Great Plains (vertical line from North Dakota to Texas) which represent some of the highest categories of rurality and some of the lowest income rates (outside of metropolitan areas in the region). Second, the error rates are higher in the Appalachian region (from southern New York to northern Mississippi) which is associated to some of the lowest income ratios in the country. Similar visualizations were observed for 1-day predictions for Model 2. Visualizations for 7-day predictions did not show clear spatial trends, possibly due to the weaker correlations reported.”

7) Comment: “The research paper mentions little of their own limitations which could affect the replicability of the study and reliability of the results. These limitations could be made clear to the reader in the discussion section. For example, although the correlations were statistically significant, certain correlations were extremely small, even below 0.1. A statistically significant correlation speaks more that the probability of such a correlation happening rather than the strength of the correlation between variables. This should be clarified to the reader. Another limitation is the potential bias in using US census data to represent demographic information. The authors did mention that COVID-19 data collection process could be biased, and this problem also extends to US census data.

• Response: Your constructive feedback is greatly appreciated. In the new limitations section, we have added a paragraph to clearly discuss limitations related to the methodology, the data, and our approach so the reader will be aware of all these important points. In lines 320-322 and 334-342, please refer to revisions that address the two concerns mentioned in this comment with respect to correlation coefficient interpretation limitations, and possible bias associated with COVID-19 case reporting and US census tract data. The edited text is also shown below. 

“While this study addressed potential biases in mobility data currently used by two types of predictive models, there are a number of limitations related to modeling and dataset biases that require clarification.”

“We have reported fairness analysis results in terms of correlation coefficients between performance and socio-economic variables. Nevertheless, statistically significant correlations reflect the probability of such a correlation occurring rather than its strength. Correlation coefficient strengths can be interpreted differently across scientific fields, and authors should avoid overinterpreting associations 60-62. As a final point, in addition to mobility data, there are other sources of data that might include biases and have an effect on prediction performance, such as diverse COVID-19 case reporting methodologies and US census tract data, both of which are used in this paper. It is important to note that these are potential biases in our study, and future work should look into their effect on the fairness analysis presented in this paper.”

8) Comment: “The interpretability of the results could be expanded. Is there a reason why the 1-day models showed a higher bias that the 7-day models? Is there a significant difference in bias between the linear regression vs. the time series forecasting?”

• Response: Thank you for your constructive comments. In the discussion section, we have added one paragraph to discuss more in depth the interpretation of our results. A higher bias rate is reported for 1-day predictions, possibly because as related work has shown, it is more difficult to predict longer time windows (7-days), leading to noisier predictions which might in turn generate lower correlations between performance and sociodemographic factors for the 7-day prediction. In addition, we have performed a Wilcoxon rank sum test between linear regression and time series forecasting coefficients reported for 1-day prediction. The results show that there is no significant statistical difference between the reported coefficients in Tables 2 and 3. Please refer to revisions in lines 291-301 and in the text below. 

“For both predictive models, we observed higher biases in 1-day predictions compared to 7-day predictions. We posit this is possibly due to the already reported difficulty in predicting covid-19 cases for higher lookaheads 56, thus resulting in noisier predictions which might in turn generate lower correlations between model performance and sociodemographic characteristics. When comparing the regression and time series models in terms of their biases, we did not observe any statistically significant difference between the reported correlation coefficients. A Wilcoxon rank sum test, also known as a Mann-Whitney U test 57, was implemented to compare two independent groups of coefficients on a 1-day prediction period (Tables 2 and 3). We chose this test because it is a non-parametric statistical test that is not based on assumptions of normality or equal variance. With a P-value of 0.289, the null hypothesis was accepted pointing to no significant difference between bias coefficients between the two models.”

Reviewer 2: 

9) Comment: “The paper reviews a prediction of COVID-19 cases based only on mobility data and raises the question of whether these predictions are reliable, considering the demographic differences among the population. The analysis is done in two stages, first, the authors run a linear regression and a time-series forecasting to predict the number of COVID-19 cases. Then, they apply a Spearman correlation analysis between the errors of the model and socio-economic and demographic characteristics. The authors concluded that in bigger regions, with highly educated, wealthy, young, and urban areas the prediction errors are smaller. Although, in general, the paper is easy to read, some parts could definitely be improved and clarified.”

• Response: 

We appreciate your positive feedback. We hope the changes that have been made address your concerns. 

10) Comment: “The main concern is the methodology choice, also, considering the low resulting correlation, the validity of the conclusions is a bit questionable. An alternative may be a choice of more sophisticated models for COVID-19 spread (for example, the ones which directly include socio-economic characteristics), to further compare their prediction errors with the regressions suggested by authors.”

• Response: Your feedback is greatly appreciated. Even though there exists a plethora of COVID-19 predictive models, we made several conscious modeling choices. We focused on linear regressions and time series due to its interpretability, as opposed to more complex models that lack interpretability (deep learning) or require training large numbers of parameters (compartmental models). We also refrained from incorporating demographic and socio-economic features into the predictive models due to its controversial nature. In fact, prior work has shown that incorporating socio-economic or demographic data into predictive models can reinforce biases and perpetuate inequalities. As a result of your comment, we have created a new limitations section where we highlight the limitations related to model selection as well as potential areas of future research for comparing models with and without demographic features as input in terms of prediction fairness. Please refer to revision in lines 343-357 or to the text below. We address the low correlation comment in the next response. 

“We made several conscious modeling choices. First, we focused on linear regressions and time series models due to its interpretability. In contrast with more complex epidemiological models that are hard to tune due to its parametric nature, and deep learning models that are difficult to interpret, linear models and time series are easy to train and test 21,28-30. Second, we have avoided incorporating socio-economic and demographic variables as input to the linear regression and time series prediction models. This choice was based on prior work showing that the addition of demographic features as input to predictive models is not only controversial but also potentially harmful 58. In fact, it has been argued that using socio-economic or demographic data as predictor may instead reinforce bias and generate predictions based primarily on demographic variables rather than on more actionable parameters, thus perpetuating inequalities 59. Future studies could consider incorporating demographic features as inputs to the predictive models to replicate the fairness analysis presented in this paper. Third, we trained individual prediction models per county. Future work should explore a unified model that learns COVID-19 trends for all counties. This approach would allow for inclusion of county-level variables indicative of population vulnerability directly in the model, potentially yielding more accurate results.”

11) Comment: “1) The resulting correlation values look quite low. Could you please elaborate on the validity of the conclusions in the studied context?”

• Response: Thank you for your comment. Most of the correlation coefficients reported are statistically significant pointing to the existence of a relationship between the error rates and various socio-economic and demographic features. We agree with the reviewer in that some of the coefficients are low, especially for the 7-day predictions, albeit statistically significant. We also acknowledge that correlation coefficient strengths can be interpreted differently across scientific fields, and authors should avoid overinterpreting the strength of the associations 59. We have addressed this comment in the paper by incorporating this limitation in the new limitations section, see lines 291-474 and the text below. In addition, we have also added a new paragraph in the discussion section where we discuss differences in correlation coefficients between the two models and between the 1-day and 7-day predictions via a new statistical analysis. Please see text in lines 334-338 and below. 

“We have reported fairness analysis results in terms of correlation coefficients between performance and socio-economic variables. Nevertheless, statistically significant correlations reflect the probability of such a correlation occurring rather than its strength. Correlation coefficient strengths can be interpreted differently across scientific fields, and authors should avoid overinterpreting associations 64-66.”

“For both predictive models, we observed higher biases in 1-day predictions compared to 7-day predictions. We posit this is possibly due to the already reported difficulty in predicting covid-19 cases for higher lookaheads 56, thus resulting in noisier predictions which might in turn generate lower correlations between model performance and sociodemographic characteristics. When comparing the regression and time series models in terms of their biases, we did not observe any statistically significant difference between the reported correlation coefficients. A Wilcoxon rank sum test, also known as a Mann-Whitney U test 57, was implemented to compare two independent groups of coefficients on a 1-day prediction period (Tables 2 and 3). We chose this test because it is a non-parametric statistical test that is not based on assumptions of normality or equal variance. With a P-value of 0.289, the null hypothesis was accepted pointing to no significant difference between bias coefficients between the two models.”

12) Comment: “2) The preparation of the datasets for the models is not clear. In particular, there is no information about the size of the dataset, its granularity, and the shares of the training and testing dataset. Also, it would be interesting to see the number of counties and their size.”

• Response: Thank you for your comment. To respond to this, we added detailed information on the size, granularity, and preparation of our datasets. Please refer to lines 101-110 (text below) and newly added Table 1 for revisions. In addition, we have added a detailed discussion to clarify training and testing, please check lines 160-173 and text below. 

“Dataset preparation

We took the following steps to prepare the final dataset for modeling. Daily mobility data was collected from the origin-destination-time (ODT) platform34 from April 14th to December 30th . The platform had daily mobility OD flows for 3,036 out of the 3,142 counties in the US. As a result, the total dataset size was of over 774,000 records. We used the Federal Information Processing Standard (FIPS) code, to match the daily number of COVID-19 cases per county with its corresponding mobility data. Therefore, the final dataset represented daily count of infections and mobility metrics per county in the US throughout the period of study. Following a similar procedure, we added socio-economic and demographic features at the county level to each data record using the FIPS code and the variables provided by the 5-year US ACS census from 2020.”

“The two models were trained at the county level on a daily basis using both COVID-19 case numbers and changes in mobility OD flows as independent variables to predict future cases. Socio-economic and demographic data were not used to train the models. That information was exclusively used during the fairness evaluation. For the linear regression model (Model 1), 21 days of mobility and past COVID-19 case data were used at a time for the training, and the trained model was used to test 1-day and 7-day predictions. We implemented a 1-day sliding window to replicate this train-test approach throughout the time period of analysis and reported average daily prediction error rates. Similarly, the time series model (Model 2), was also trained using a typical training-testing window approach for time series predictions 48, with a 90-day training dataset. Using a 1-day sliding window on the training dataset, this approach resulted in predictions available from early August to the end of the year. Different training lengths were evaluated for both models, and the ones with the best accuracies were selected. In this process, thousands of regressions and ARIMAX models are trained at the county level on a daily basis to be able to predict COVID-19 cases.”

13) Comment: “3) It is not clear how the input of linear regression was formed – were all counties considered together or the coefficients were calculated separately for each county? It would be also convenient to summarize the explanatory variables fed to each model.”

• Response: Thanks for the constructive comment. As described in the previous response, we have further clarified the training and testing approaches for both models in the revised draft. The two models are trained at the county and daily levels separately. The inputs (independent variables) for both models were exclusively: (1) changes in prior mobility OD flows and (2) the number of past COVID-19 cases. No socio-economic or demographic data was used during the model training. Instead that data was used for the fairness analysis. Please refer to revision in lines 160-173. We have also added a clarification in the limitations section mentioning our focus on individual county models, and proposing a unified model for future work. Please refer to revision lines 354-357, and text below. 

“The two models were trained at the county level on a daily basis using both COVID-19 case numbers and changes in mobility OD flows as independent variables to predict future cases. Socio-economic and demographic data were not used to train the models. That information was exclusively used during the fairness evaluation. For the linear regression model (Model 1), 21 days of mobility and past COVID-19 case data were used at a time for the training, and the trained model was used to test 1-day and 7-day predictions. We implemented a 1-day sliding window to replicate this train-test approach throughout the time period of analysis and reported average daily prediction error rates. Similarly, the time series model (Model 2), was also trained using a typical training-testing window approach for time series predictions 48, with a 90-day training dataset. Using a 1-day sliding window on the training dataset, this approach resulted in predictions available from early August to the end of the year. Different training lengths were evaluated for both models, and the ones with the best accuracies were selected. In this process, thousands of regressions and ARIMAX models are trained at the county level on a daily basis to be able to predict COVID-19 cases.”

“We trained individual prediction models per county. Future work should explore a unified model that learns COVID-19 trends for all counties. This approach would allow for inclusion of county-level variables indicative of population vulnerability directly in the model, potentially yielding more accurate results. “

14 Comment: The abbreviations (for example, NCHS at line 192) and variable in equations should be defined.

• Response: Thanks for pointing this out. We have added the NCHS abbreviation to line 105. NCHS stands for National Center for Health Statistics Urban-Rural Classification. The paper was also checked for other abbreviations. Additionally, we have further clarified all variables in equations (1) and (2). Please review revisions in lines 131-133 and 144-150.

---

## [Decision Letter · Decision Letter 1]

28 Jul 2023

PONE-D-23-04556R1A fairness assessment of mobility-based COVID-19 case prediction modelsPLOS ONE

Dear Dr. Erfani,

Thank you for submitting your manuscript to PLOS ONE. After careful consideration, we feel that it has merit but does not fully meet PLOS ONE’s publication criteria as it currently stands. Therefore, we invite you to submit a revised version of the manuscript that addresses the points raised during the review process. Both reviewers suggest that a minor revision of the manuscript may be sufficient for publication of the work. Please revise your manuscript according to the received recommendations, being careful to highlight the changes in the manuscript.

We look forward to receiving your revised manuscript.

Kind regards,

Emanuele Crisostomi, PhD

Academic Editor

PLOS ONE

Journal Requirements:

Reviewers' comments:

Reviewer's Responses to Questions

**Comments to the Author**

1. If the authors have adequately addressed your comments raised in a previous round of review and you feel that this manuscript is now acceptable for publication, you may indicate that here to bypass the “Comments to the Author” section, enter your conflict of interest statement in the “Confidential to Editor” section, and submit your "Accept" recommendation.

Reviewer #1: (No Response)

Reviewer #2: (No Response)

2. Is the manuscript technically sound, and do the data support the conclusions?

Reviewer #1: Yes

Reviewer #2: Partly

3. Has the statistical analysis been performed appropriately and rigorously? 

Reviewer #1: Yes

Reviewer #2: Yes

4. Have the authors made all data underlying the findings in their manuscript fully available?

Reviewer #1: Yes

Reviewer #2: Yes

5. Is the manuscript presented in an intelligible fashion and written in standard English?

Reviewer #1: Yes

Reviewer #2: Yes

6. Review Comments to the Author

Reviewer #1: - Thank you to the authors for addressing these comments. I appreciate that the authors carefully addressed the limitations of their research. They also included choropleth maps and an interesting discussion on the error prediction rates from a spatial perspective. The authors also added an analysis of the difference between 1-day and 7-day predictions. I recommend a minor revision of the journal to adjust some formatting and grammar issues. Please address these minor comments:

-

- 1. In Table 1, please include the appropriate unit of measurement either in the description or the individual cells. For example, Income could be” Median household income ($) “ or “$70, 264.80”. I also observed that all the variables are at the county level, so this does not need to be repeated and should instead be in the Table description.

- 2. In Table 1, the description for smartphones is “percentage of the population at the county level with a bachelor degree and above.” Please correct this typo.

- 3. In Figure 5, can you label which error map is Model 1 and Model 2? Also, I notice that there were more maps in the comment response that were not in the manuscript. Were these moved to supplemental information? If so, this should be referenced in the manuscript.

- 4. I recommend that the authors review the grammar of the manuscript once more. There is an odd use of punctuation and run-on sentences which could be revised. I provided some examples but please review the whole manuscript.

E.g. “These results reveal that Model 2 – an ARIMAX with mobility data added as exogenous variable – performs better (i.e., has lower errors) in counties that share higher income, higher smartphone ownership, larger populations, and higher educational levels (see Figure 4 for weekly representations of the weekly correlations for some of these features).”

“Given that these findings are replicated across models 1 and 2, thus controlling for algorithmic bias, we posit that the model is unfair in part due to the bias in the mobility data used in the model, although bias in the way COVID-19 case data is collected, could also influence the outcome.”

Reviewer #2: The authors addressed the majority of my comments; however, I still believe that they should better clarify how they interpret the correlation coefficients.

In particular, it is stated that “Correlation coefficient strengths can be interpreted differently across scientific fields, and authors should avoid overinterpreting associations”. As the main conclusion of the work is based on these correlation coefficients, it would be helpful to clarify how to interpret the obtained values in the context of their paper.

Since different interpretations of the correlation coefficient exist, the authors should better clarify the rationale of their choice and how they interpret those results. Specifically, they should better describe how the values in the table should be interpreted and avoid qualitative statements.

In addition, an extended description of the results should be added. Which sociodemographic features are more likely to cause bias, and which are almost negligible? Why does the correlation vary so much among different months (for example, in Table 2 the Education in April is -0.25 and in December -0.06)?

Also, references to the tables (for example, line 248) and the content of Table 1 should be checked.

7. PLOS authors have the option to publish the peer review history of their article (what does this mean?). If published, this will include your full peer review and any attached files.

Reviewer #1: No

Reviewer #2: No

---

## [Author Response · Author response to Decision Letter 1]

24 Aug 2023

The authors are grateful for the opportunity to resubmit the article. Please accept our thanks for the time and effort that the editors and reviewers spent reviewing our manuscript. All comments were seriously considered, and we conducted a thorough revision of the entire paper according to reviewers’ feedback. We hope the changes listed have made the manuscript suitable for publication and we look forward to your response. 

Please note that due to the changes, line and page numbers referenced in the reviewers’ comments might be different compared to the updated version. Where applicable, page and line numbers are listed based on the new version which can be tracked.

Editor: 

1) Comment: “Thank you for submitting your manuscript to PLOS ONE. After careful consideration, we feel that it has merit but does not fully meet PLOS ONE’s publication criteria as it currently stands. Therefore, we invite you to submit a revised version of the manuscript that addresses the points raised during the review process.

Both reviewers suggest that a minor revision of the manuscript may be sufficient for publication of the work. Please revise your manuscript according to the received recommendations, being careful to highlight the changes in the manuscript.”

• Response: We appreciate the time spent reviewing our paper by the department editor, and the reviewers. Our paper has been revised in response to your feedback. Please see below for our detailed responses to each particular comment. 

Reviewer 1: 

2) Comment: “Thank you to the authors for addressing these comments. I appreciate that the authors carefully addressed the limitations of their research. They also included choropleth maps and an interesting discussion on the error prediction rates from a spatial perspective. The authors also added an analysis of the difference between 1-day and 7-day predictions. I recommend a minor revision of the journal to adjust some formatting and grammar issues. Please address these minor comments:”

• Response: We appreciate your constructive comments. We hope the changes we have made have enhanced our paper to make it ready for publication. 

3) Comment: “- 1. In Table 1, please include the appropriate unit of measurement either in the description or the individual cells. For example, Income could be” Median household income ($) “or “$70, 264.80”. I also observed that all the variables are at the county level, so this does not need to be repeated and should instead be in the Table description.

• Response: Thanks for your suggestions. We revised the table format accordingly. 

4) Comment: “- 2. In Table 1, the description for smartphones is “percentage of the population at the county level with a bachelor degree and above.” Please correct this typo.”

• Response: Thank you for your comment. We have corrected the typo. 

5) Comment: “- 3. In Figure 5, can you label which error map is Model 1 and Model 2? Also, I notice that there were more maps in the comment response that were not in the manuscript. Were these moved to supplemental information? If so, this should be referenced in the manuscript.”

• Response: Thank you very much for your comment. We have added the model number in the caption of Figure 5. In addition, we have created a supplemental materials file where we have added the remaining figures. As suggested by the reviewer, we have also added a reference to the supplemental materials in the text. 

6) Comment: “- 4. I recommend that the authors review the grammar of the manuscript once more. There is an odd use of punctuation and run-on sentences which could be revised. I provided some examples but please review the whole manuscript.

E.g. “These results reveal that Model 2 – an ARIMAX with mobility data added as exogenous variable – performs better (i.e., has lower errors) in counties that share higher income, higher smartphone ownership, larger populations, and higher educational levels (see Figure 4 for weekly representations of the weekly correlations for some of these features).”

“Given that these findings are replicated across models 1 and 2, thus controlling for algorithmic bias, we posit that the model is unfair in part due to the bias in the mobility data used in the model, although bias in the way COVID-19 case data is collected, could also influence the outcome.”

Response: Thank you for your feedback. We have proofread the paper for grammatical and writing errors.

Reviewer 2: 

7) Comment: “The authors addressed the majority of my comments; however, I still believe that they should better clarify how they interpret the correlation coefficients.

In particular, it is stated that “Correlation coefficient strengths can be interpreted differently across scientific fields, and authors should avoid overinterpreting associations”. As the main conclusion of the work is based on these correlation coefficients, it would be helpful to clarify how to interpret the obtained values in the context of their paper.

Since different interpretations of the correlation coefficient exist, the authors should better clarify the rationale of their choice and how they interpret those results. Specifically, they should better describe how the values in the table should be interpreted and avoid qualitative statements.”

• Response: Thank you for your comment. We mostly focused our fairness analysis and discussion on the significance and direction of the results i.e., p-values and positive or negative correlation signs. Following this approach, our results show statistically significant differences with all p-values under 0.05 that is, with at least 95% confidence in the results. We did not discuss strength in depth because as we state in the limitations section, coefficient strengths can be interpreted differently across scientific fields, and there is no formal agreement on how to define correlation strength for covid-19 studies. Nevertheless, we agree with the reviewer that our interpretation approach needs to be clearer to strengthen our analysis. Thus, we researched papers using spearman rank correlation in the contexts of medicine and big data analysis and agreed to select the coefficient of 0.3 as the threshold between high and low correlation (notation used in paper 2, see information below). To describe the strength of the correlation, we will use the words weak (below 0.3) and moderate (above 0.3) as defined in paper 1, see information below. We have also updated the results section accordingly to include a brief analysis of the correlation strength using the coefficient threshold selected based on existing work. We have also added language in the results section to clarify that the strength of the statistically significant relationship is low (weak). Finally, we have clarified our threshold selection in both the Methods and the Limitations sections. See edited text below. 

In the Methods section:

“To discuss correlation strength, and based on prior work, we will use 0.3 as the correlation coefficient threshold between a high and low correlation 51, or a weak and a moderate correlation 52.”

In the Limitations Section:

“Based on prior work utilizing spearman rank correlation in the context of medicine and big data analysis, we have selected a correlation coefficient of 0.3 as the threshold between high and low correlation 51, or weak and moderate correlation 52.” 

Paper 1: Akoglu, Haldun. "User's guide to correlation coefficients." Turkish journal of emergency medicine 18, no. 3 (2018): 91-93.

Paper 2: Xiao, Chengwei, Jiaqi Ye, Rui Máximo Esteves, and Chunming Rong. "Using Spearman's correlation coefficients for exploratory data analysis on big dataset." Concurrency and Computation: Practice and Experience 28, no. 14 (2016): 3866-3878.

8) Comment: “In addition, an extended description of the results should be added. Which sociodemographic features are more likely to cause bias, and which are almost negligible?”

Response: Thank you for your constructive comment. As mentioned in our prior response, we have focused the discussion of our results on both significance and direction of the coefficients i.e., p-values and positive or negative correlation signs. Given that the strength of the correlations found is weak, we posit that all socio-economic and demographic features are related to significant, albeit weak, bias. We do not observe statistically significant differences in the strengths across features. We do however see that correlation coefficients are smaller for 7-day predictions than 1-day predictions, which might point to more negligible bias for these models. Nevertheless, the statistical results do not support any feature being more biased than other. We have clarified this in the results discussion. See text below. 

 “Given that the strength of the correlations found is weak, we posit that all socio-economic and demographic features are related to significant, albeit weak, bias. We do not observe statistically significant differences in the strengths across features. We do however see that correlation coefficients are smaller for 7-day predictions than 1-day predictions, which might point to more negligible bias for these models. Nevertheless, the statistical results do not support any feature being more biased than other.”

9) Comment: “Why does the correlation vary so much among different months (for example, in Table 2 the Education in April is -0.25 and in December -0.06)?”

• Response: 

We appreciate your feedback. As presented in related work (paper 1 and 2 below), earlier pandemic months showed strong, uniform mobility changes across the population, with large percentages of individuals reducing their mobility. We posit that these widespread changes make it easier to find associations (correlations) between mobility behaviors and other socio-economic and demographic data. However, during later pandemic months, and especially once vaccines were made available (from December onwards), mobility behaviors were more entropic, with population groups showing very diverse mobility behaviors across the US. As papers 1 and 2 show, these non-uniform changes made mobility data less useful in the analysis of covid-19 settings. We posit that these non-uniform changes make it harder for the analysis to find any associations or patterns of behavior between mobility and socio-economic and demographic data thus making the correlations much weaker. We have edited to results section to include this clarification. See text below. 

“It is important to highlight that for certain socio-economic and demographic features in Tables 2 and 3, we observe some variance in the correlation coefficients across months, with earlier pandemic months showing higher coefficients. We posit that these might be due to mobility behaviors being more entropic later during the pandemic, which might make it harder to find associations. As prior work has shown, uniform mobility behaviors made mobility data more useful in predictive models at the onset of the pandemic than in later periods 20,53.”

Paper 1: Badr, Hamada S., and Lauren M. Gardner. "Limitations of using mobile phone data to model COVID-19 transmission in the USA." The Lancet Infectious Diseases 21, no. 5 (2021): e113.

Paper 2: Gatalo, Oliver, Katie Tseng, Alisa Hamilton, Gary Lin, and Eili Klein. "Associations between phone mobility data and COVID-19 cases." The Lancet Infectious Diseases 21, no. 5 (2021): e111.

10) Comment: “Also, references to the tables (for example, line 248) and the content of Table 1 should be checked.”

• Response: Your feedback is greatly appreciated. We revised the typo in table referencing and in Table 1 content.

---

## [Decision Letter · Decision Letter 2]

12 Sep 2023

A fairness assessment of mobility-based COVID-19 case prediction models

PONE-D-23-04556R2

Dear Dr. Erfani,

We’re pleased to inform you that your manuscript has been judged scientifically suitable for publication and will be formally accepted for publication once it meets all outstanding technical requirements.

Kind regards,

Emanuele Crisostomi, PhD

Academic Editor

PLOS ONE

Additional Editor Comments (optional):

Reviewers' comments:

Reviewer's Responses to Questions

**Comments to the Author**

1. If the authors have adequately addressed your comments raised in a previous round of review and you feel that this manuscript is now acceptable for publication, you may indicate that here to bypass the “Comments to the Author” section, enter your conflict of interest statement in the “Confidential to Editor” section, and submit your "Accept" recommendation.

Reviewer #1: All comments have been addressed

Reviewer #2: All comments have been addressed

2. Is the manuscript technically sound, and do the data support the conclusions?

Reviewer #1: Yes

Reviewer #2: (No Response)

3. Has the statistical analysis been performed appropriately and rigorously? 

Reviewer #1: Yes

Reviewer #2: (No Response)

4. Have the authors made all data underlying the findings in their manuscript fully available?

Reviewer #1: Yes

Reviewer #2: (No Response)

5. Is the manuscript presented in an intelligible fashion and written in standard English?

Reviewer #1: Yes

Reviewer #2: (No Response)

6. Review Comments to the Author

Reviewer #1: All comments have been fully addressed. Thank you to the authors for their work in improving the manuscript

Reviewer #2: The authors satisfactorily addressed all my comments. I suggest accepting the paper in its current form.

7. PLOS authors have the option to publish the peer review history of their article (what does this mean?). If published, this will include your full peer review and any attached files.

Reviewer #1: No

Reviewer #2: No

---

## [Editor Report · Acceptance letter]

10 Oct 2023

PONE-D-23-04556R2 

A fairness assessment of mobility-based COVID-19 case prediction models 

Dear Dr. Erfani:

I'm pleased to inform you that your manuscript has been deemed suitable for publication in PLOS ONE. Congratulations! Your manuscript is now with our production department. 

Kind regards, 

on behalf of

Professor Emanuele Crisostomi 

Academic Editor

PLOS ONE